# Subsurface fracturing of sedimentary stones caused by bullet impacts

Oliver Campbell[1]*, Tom Blenkinsop[1], Lisa Mol[2], Oscar Gilbert[2]

1 School of Earth and Ocean Sciences, Cardiff University, Cardiff, United Kingdom, 2 Department of Geography and Environmental Management, University of the West of England, Bristol, United Kingdom

* campbellor@cardiff.ac.uk

## Abstract

The immovable nature of built heritage means that it is particularly vulnerable during times of armed conflict. Although impacts from small arms and shrapnel leave relatively inconspicuous impact scars, they elevate the risk of future stone deterioration. This study investigates the subsurface damage caused by bullet impacts, which is not apparent from surface inspection, in order to better understand the geometry and mechanics of this form of conflict damage to heritage. Controlled firearm experiments were conducted to simulate conflict damage to sandstone and limestone buildings. The bullet impacts created conical fractures or zones of increased fracture intensity below the impact, radial fractures, and spallation, in addition to a crater. Dynamic fracture distinguishes the formation of these features from quasi static cone crack experiments, while the lack of a shockwave differentiates these bullet impacts from hypervelocity experiments. Damage was created by momentum transfer from the bullet, so that differences in target properties had large effects on the nature of the damage. The crater in the limestone target was almost an order of magnitude deeper than the sandstone crater, and large open fractures formed in the limestone below the crater floor, compared with zones of increased fracture intensity in the sandstone target. Microstructural analysis of subsurface damage showed that fracture intensity decreased with increasing distance from the impact centre, suggesting that regions proximal to the impact are at increased risk of future deterioration. Conical subsurface fractures dipping away from the impact beneath multiple impact craters could link up, creating a continuous fracture network. By providing pathways for moisture and other weathering agents, fractures enlarge the region at increased risk of deterioration. Their lack of surface expression makes understanding their formation a vital part of future surveying and post conflict assessments.

## Introduction

The recent invasion of Ukraine has brought the damage and destruction caused by modern weaponry to the forefront of public attention. Long range artillery and missiles cause significant destruction to their targets, and shrapnel generated in explosions can damage surrounding structures. Bullet impacts from small arms add further damage to buildings and monuments, especially during urban firefights. Russian advances into Kyiv's western suburbs

**Data Availability Statement:** All relevant data are within the manuscript and its Supporting information files.

**Funding:** This research was funded by the Leverhulme Trust, grant number RPG-2017-408.

The APC was funded by Cardiff University's Institutional Open Access Fund. The funders had no role in study design, data collection and analysis, decision to publish, or preparation of the manuscript.

**Competing interests:** The authors have declared that no competing interests exist.

of Irpin, Bucha, and Hostomel in late February 2022 led to urban tank and infantry battles, damaging multiple heritage sites and buildings (Fig 1) [1, 2].

There are very few detailed studies of the surface damage caused by bullet and shrapnel impacts, and its relationships to the subsurface. In a study of bullet and shrapnel impacts to limestone walls and window ledges, Mol and Gomez-Heras [3] observed lower surface hardness measurements in the regions surrounding impact craters and fractures than in areas of undamaged stone. Ultra-pulse velocity measurements suggested an increase in subsurface fractures in regions proximal to the surficial impacts [3]. A controlled impact study by Gilbert et al. [4] found similar reduced surface hardness near the surface crater caused by a bullet impact, as well as spatial correlations between increased surface permeability measurements, surface fractures, and impact craters. Microstructural analysis of the same sandstone sample found grain crushing at the floor of the impact crater, as well as intra- and intergranular fracturing [5]. Subsurface imaging from thin sections showed fractures had a mix of inter- and intragranular pathways close to the crater floor, becoming predominantly intergranular with increasing distance from the crater floor, with fracture intensity decreasing with increasing distance from the crater centre [5]. These studies show that extensive subsurface damage can occur from bullet impacts, which is not readily appreciated from the surface effects. However, details of subsurface damage from bullet impacts, and particularly the mechanisms that cause it, are not known.

Fracturing within a rock mass reduces its overall strength, increases its effective porosity, and can act as conduits for moisture ingress [6–8]. Moisture can dissolve constituent grains and/or cement in sedimentary rocks, widening pore spaces and further decreasing overall rock strength, exacerbating a negative feedback loop of stone deterioration. Moisture also

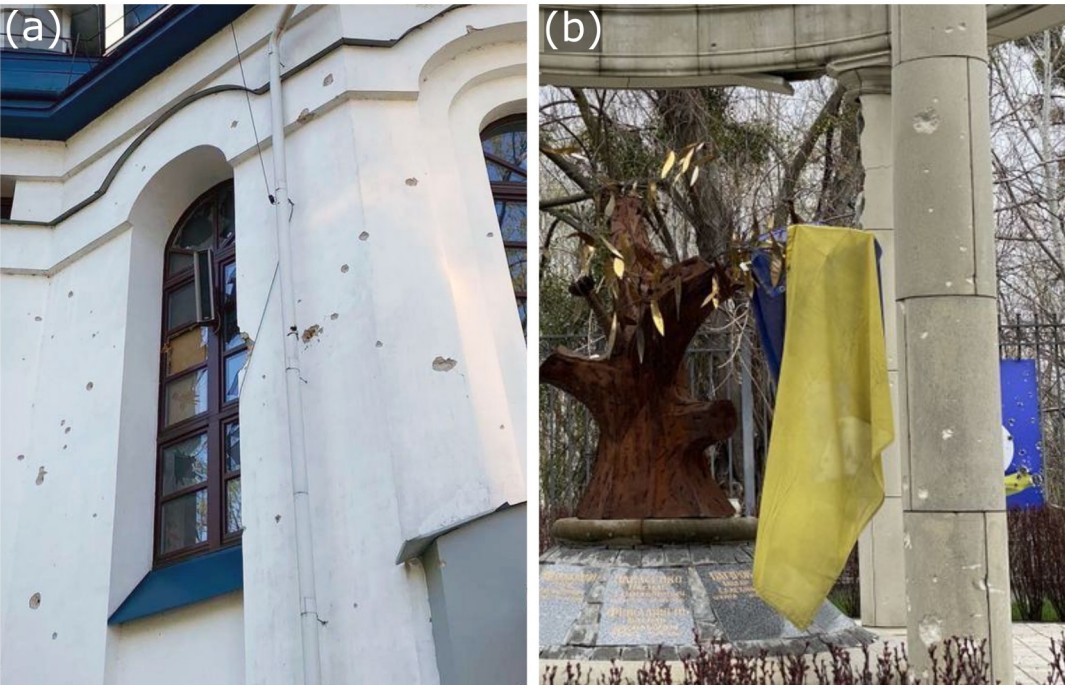

**Fig 1. (a)** Shrapnel damage to the facade of the St Nicholas Church caused by Russian shelling in the town of Irpin, a suburb to the NW of Kyiv in Northern Ukraine [2]. **(b)** Impact damage to columns of the Alley of ATO Heroes memorial, also in Irpin. It is reported to have been caused by small arms fire from Russian forces in February 2022 [1, 2].

transports dissolved salts, which apply an outward pressure upon crystallisation, weakening cement-grain boundaries and the cohesiveness of the stone, resulting in material loss from the surface of the stone over time [9–13]. Increased fracture intensity enhances the progression of weathering fronts in granitic rocks [14]. Other fracture characteristics, such as aperture, orientation, and connectivity, influence stone permeability and the flow of fluids [15]. A thorough characterisation of internal damage caused by bullet impacts is therefore important for understanding the vulnerability of stone to weathering processes and deterioration.

This study aims to characterise and quantify the subsurface damage caused by modern rifle bullets in two sedimentary stone types, to understand the damage mechanisms, and to link the damage to potential deterioration of built heritage. Observations of fracture morphology in thin sections are combined with fracture intensity analysis of digitised fracture maps to examine how subsurface damage changes with distance to the crater centre.

## Methods and materials

### Impact experiments

Freshly quarried cubes (15 x 15 x 15 cm) of Stoneraise Red Sandstone (SRS) and Cotswold Hill Cream Limestone (CHCL) were selected as the target lithologies because they are broadly representative of sandstones and oolitic limestones used for construction. SRS is a fine-medium (0.125–0.5 mm), quartz rich sandstone from the Permian New Red Sandstones (quarried near Penrith, U.K). With a porosity of 11% and intergranular cement comprising quartz overgrowths, it is generally massive, with some target blocks exhibiting visible beds of coarser grains (~1 mm) (Fig 2a). Target blocks have an average uniaxial compressive strength perpendicular and parallel to bedding of 40.0 ± 5.9 MPa and 45.0 ± 13.1 MPa respectively [16]. The average indirect tensile strength parallel to bedding (loading direction perpendicular to bedding) measured via Brazil disc tests is 5.0 ± 0.3 MPa [16]. CHCL is an oolitic grainstone from the Middle Jurassic Inferior Oolite (quarried near Ford, U.K.). The average grain size is 0.5 mm and it has a porosity of ~20% (Fig 2b). The majority of intergranular cement comprises sparry calcite, though areas of smaller grain sizes havemicrocrystalline calcite cement. Target blocks have an average uniaxial compressive strength perpendicular and parallel to bedding of 10.6 ± 1.5 MPa and 8.8 ± 2.1 MPa respectively [16]. The average indirect tensile strength parallel to bedding (loading direction perpendicular to bedding) measured via Brazil disc tests is 2.2 ± 0.2 MPa [16]. Thin section micrographs from undamaged samples of each lithology show no pre-existing fractures (Fig 2). We can therefore be confident that all damage reported is the result of bullet impacts and not inherited.

Controlled firearm experiments were carried out at Cranfield Ordnance Test and Evaluation Centre (Gore Cross, UK) to simulate conflict damage to stone. 7.62 x 39 mm (abbreviated in this study as AK-47) is a commonly used ammunition cartridge fired from AK-variant rifles, such as the widely known AK-47 and has been used in contemporary and past conflicts. Shots were fired from a fixed proof barrel at incident angles of 90˚ to the target face. The AK-47 projectile has a spitzer ogive nose shape and is comprised of a brass jacket and lead core weighing 7.95 grams (123 grains). Propellant loads for each cartridge were adjusted to reduce velocity and simulate impacts at distances of 200 m (532 ms$^{-1}$ for the impact into the CHCL sample and 539 ms$^{-1}$ for the impact into SRS). Average engagement distances in urban firefights during the Iraq War ranged from 26 m to over 126 m between combatants, and most soldiers are trained for engagement distances of 0–600 m, so 200 m represents a reasonable distance for simulating impacts in both urban and open scenarios [17, 18]. The kinetic energy ($K_e = 1/2mv_i^2$) of the projectile at impact was ~1125 J for the CHCL experiment and ~1154 J for the SRS experiment. Concrete blocks were placed on all faces, except the target face, for

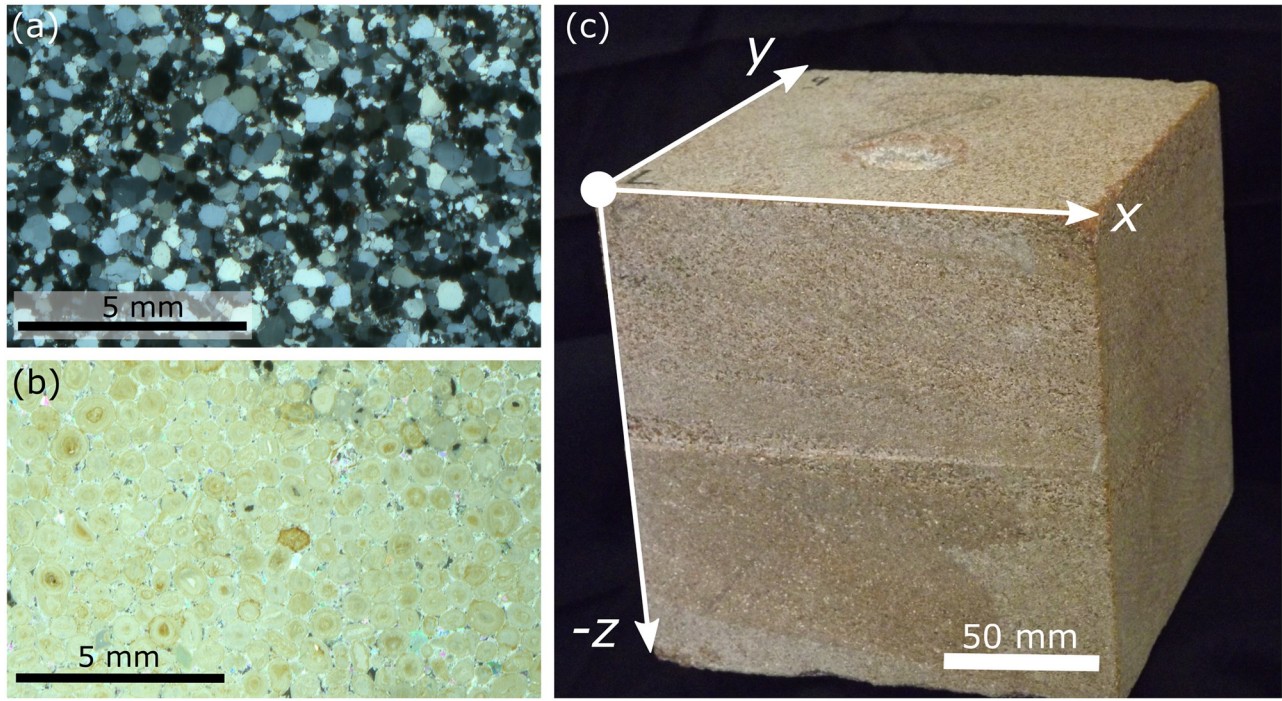

**Fig 2.** Thin section micrographs taken under cross polarised light of undamaged Stoneraise Red Sandstone under cross polarised light (a), a fine-medium grained (0.125–0.5 mm) quartz rich sandstone, and undamaged Cotswold Hill Cream Limestone (b), an oolitic limestone with an average grain size of 0.5 mm. Both lithologies show no pre-existing fracturing in the undamaged section. (c) Damaged target block of Stoneraise Red Sandstone indicating the reference scheme adapted from [19].

confinement. Target blocks with bedding were oriented so that any bedding planes present were parallel to the target face (*XY* plane).

## Microstructural damage

A 3D reference scheme, adapted from Tikoff et al., [19], was employed to describe the spatial position of thin sections, observations, and measurements within the block. The target face of the sample is defined as the *XY* plane and the *Z* axis is orthogonal to this and negative into the block (Fig 2c). The crater centre is defined as the point at the centre of the crater floor, typically the deepest point, and is used as the reference location from which to measure distances to fractures and damage within the sample.

Polished thin sections were cut from one damaged sample of SRS and one damaged sample of CHCL parallel to the *XZ* plane and transecting the centre of the crater. The target block was impregnated with epoxy resin prior to sectioning in order to minimise further damage to the samples. A combination of large (75 x 50 mm) and small sections (28 x 48 mm) were cut to maximise the coverage of impact related damage. Thin sections were scanned using an Epson Perfection 3170 photo scanner at 6400 dpi under plane and cross polarised light. Reflected light photomicrographs of each section were taken at x1 magnification using a Leica DM750P optical microscope fitted with a MC190HD camera. Microsoft ICE (Image Composite Editor) (version 2.0.3.0) was used to create a photo-mosaic of full sections. Complete photo-mosaics and thin section scans were georeferenced and fractures manually digitised in QGIS. Closed fractures were digitised as a single polyline and open fractures as a polygon to create a complete fracture map. Closed fractures are defined as fractures that, at the scale of observation, do

**Table 1. Summary of the uncertainty values for fracture intensity measurements from Stoneraise Red Sandstone (SRS) and Cotswold Hill Cream Limestone (CHCL) target lithologies.**

| Sample | Max Uncertainties (mm$^{-1}$) | | Average Uncertainty (mm$^{-1}$) |
|---|---|---|---|
| SRS_09 | - 0.0101 | + 0.0720 | + 0.0004 |
| CHCL_09 | - 0.0178 | +0.018 | + 0.0005 |

not have a distinguishable aperture. Some thin sections were subject to material loss during section production, though every effort was made to prevent this by pre-sectioning application of epoxy resin. These regions were digitised and removed from the sampling area of later analyses. The fracture map was thresholded into a binary image and the automatic fracture digitisation tool of NetworkGT (a QGIS plugin) used to generate a fracture network of polylines for analysis. This automatic digitisation approach ensures a consistent interpretation of fracture geometries and fracture characteristics across samples.

Different methods can result in varying values for important characteristics of fracture networks, such as length and orientation [20, 21]. Analysing fracture branches instead of full traces reduces this bias, as well as mitigating any censoring effects of the sample region because intersection with the edge of the sampling area affects only a single branch, instead of a full fracture trace [20]. A sample grid of systematically spaced points 0.25 mm apart, each with a sampling radius of 0.75 mm, was created within the outlines of each thin section, excluding areas of material lost during section production.

*Pxy* values provide a useful measure of fracture damage that can be compared between lithologies. *Pxy* values characterise fracture frequency, intensity, and volume, depending on the dimensions analysed. *x* represents the dimension of the sampling region and *y* the dimension of measurement [22, 23]. For example, $P_{21}$ is a measure of fracture length (*L*) per area (*A*):

$$P_{21} = \Sigma L / A \qquad (1)$$

Uncertainty in the distance from the crater centre measurements is estimated to be ± 2 mm, which combined with the uncertainty in the digitisation of fracture networks, results in the fracture intensity uncertainties presented in Table 1. A full description of uncertainty methodology is available in S1 Appendix. Fracture orientations are weighted based on fracture length and presented on equal area rose diagrams.

## Results

### Sandstone target

The sandstone sample (SRS_09) has a shallow, bowl shaped crater with an area equivalent diameter of 40 mm and a maximum depth of 5.1 mm [24]. 20 mm directly below the crater floor is an open (<1.5 mm) fracture that is 16 mm in length, but does not reach the edge of the section (Fig 3a). 80 mm directly below the crater centre there is an open fracture with a minimum aperture of 1.4 mm. Maximum aperture cannot be determined because the upper fracture wall shows evidence of material loss from sectioning (Fig 3a and 3b). Both of these open fractures are sub-parallel to the orientation of beds defined by grain size changes, ~5˚ from the target face (*XY* plane i.e. 90˚/270˚ relative to the *Z* axis in the thin sections). Two dominant orientations of fractures become apparent from the rose diagram: the first are, as described above, sub-parallel to the bedding orientation of 90˚/270˚, while the second group is approximately orthogonal to this, with orientations 0˚/180˚ (Fig 3c).

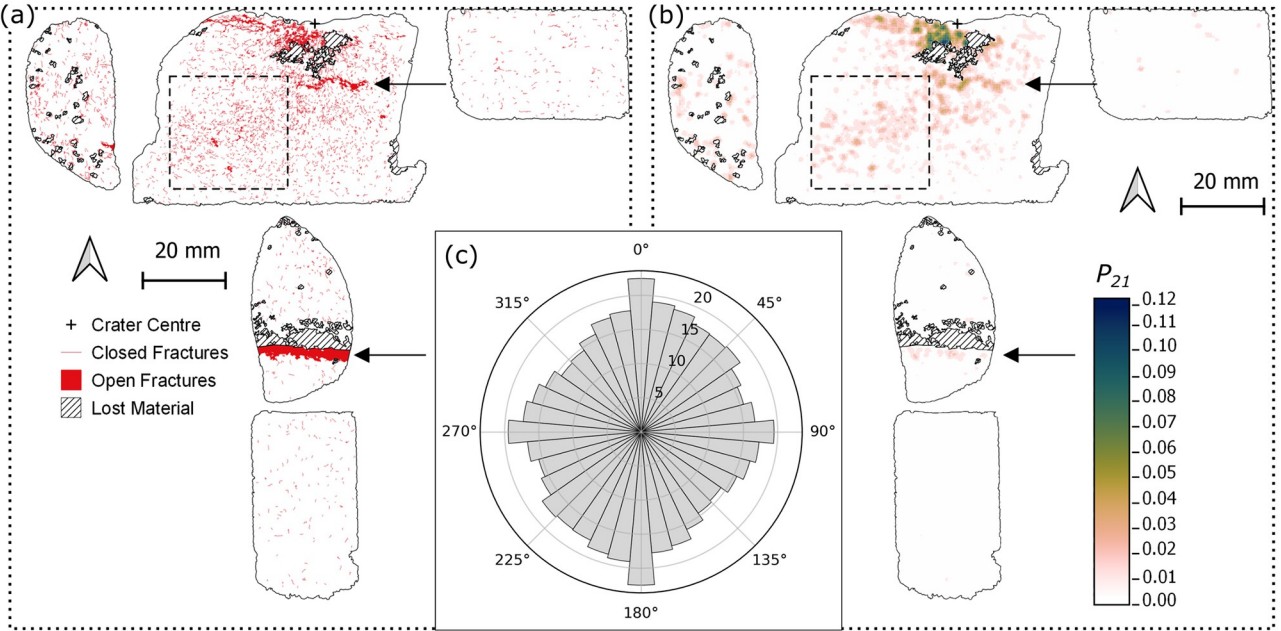

**Fig 3.** Fracture map (a) through the centre plane (XZ) of a Stoneraise Red Sandstone sample (SRS_09). Open fractures (solid red regions) are visible oriented sub-parallel to the target surface close to the impact crater, at a depth of 20 mm and ~80 mm below the crater (black arrows). There is a high number of closed (red line) fractures within a 7 mm radius of the crater centre. (b) Map of $P_{21}$ fracture intensity values across the thin sections. The highest values (dark blue) are within 7 mm of the crater centre. There is a region of relatively higher fracture intensity (dashed square) with an approximate orientation of 35°/215°. For both maps impact direction is top to bottom and the original block outline is shown as a dotted line. (c) Equal area rose diagram showing the orientation of all fractures, weighted for fracture length, mapped within the sandstone sample. Radial scale is the square root of frequency.

Directly below the crater centre is a zone of primarily closed intra- and trans-granular fractures, forming a region of intense fracturing that extends to a depth of ~ 7 mm below the crater floor (Fig 4a–4e). The highest $P_{21}$ fracture intensity value calculated (0.124) is in this region, 5.9 mm away from the crater centre (Figs 3b and 5b). Many grains exhibit multiple closed fractures that originate at contact points with adjacent grains, forming connected networks across multiple grains. Open extensional fractures are visible just beneath the crater floor traversing from the crater centre towards the rim (Fig 4f and 4g). These fractures have both inter- and trans-granular pathways, with no measurable lateral displacement between fracture walls. They are primarily sub-parallel to the target face of the samples (Fig 4h). In the top central section there appears to be a band of damage stretching from the SW corner of the section to an area of material loss directly below the crater centre (Fig 4a and 4i–4k). The band has an approximate orientation of 35°/215°.

There are few fractures in the thin sections further than 80 mm below the crater floor, and those present are short, intra-granular fractures, typically confined to a single grain. This is visible in the small peak in $P_{21}$ intensity at 80 mm below the crater centre (the large open fracture), followed by very low intensity values with increasing distance from the crater centre (Fig 5a).

## Limestone target

The limestone target (CHCL_09) has a wider (101.9 mm) and deeper (42.5 mm) crater than the sandstone sample (SRS_09) [24]. The crater has a two-part structure of a shallow, gently

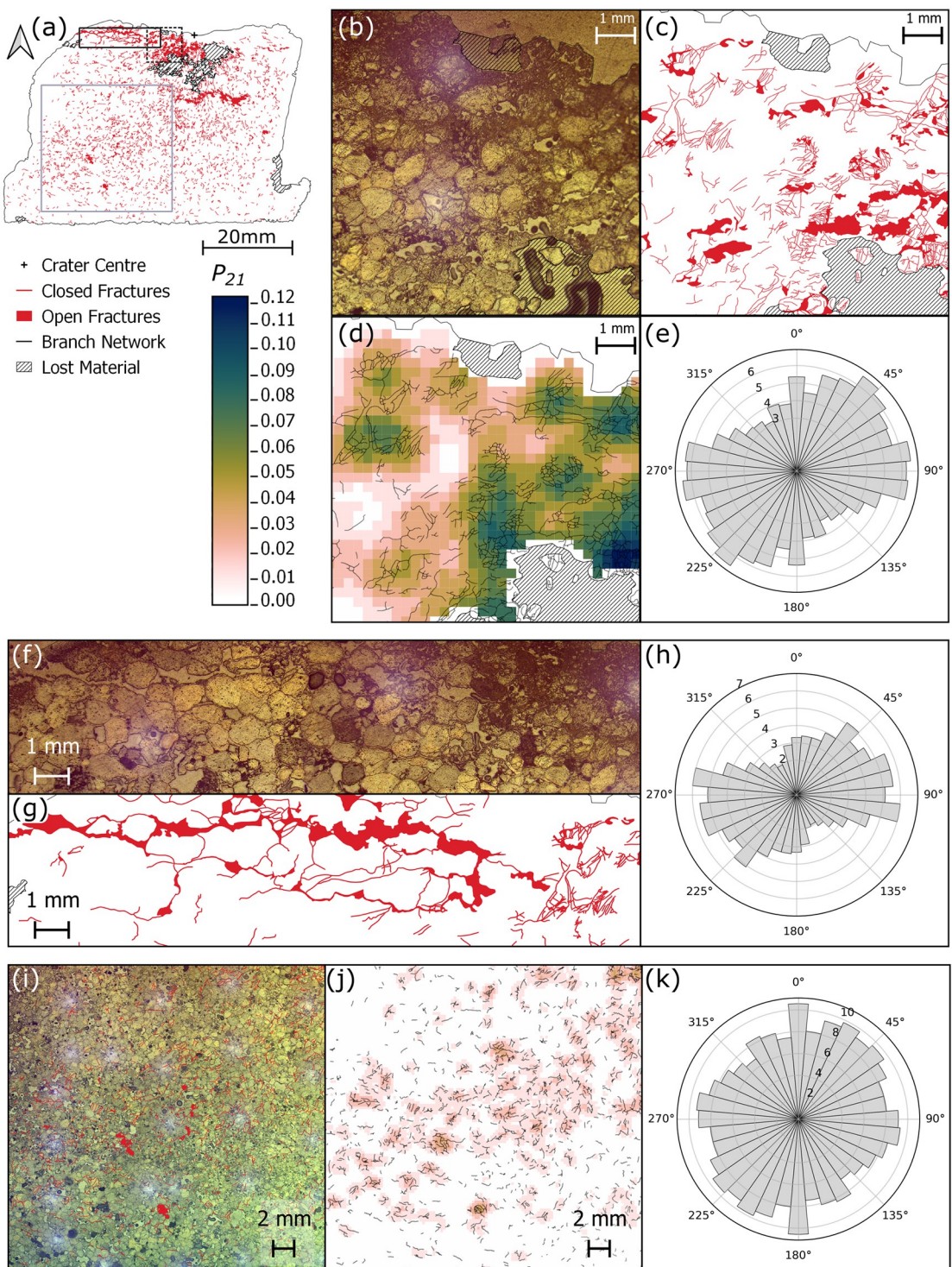

**Fig 4.** (a) Fracture map of the thin section through the impact crater in Stoneraise Red Sandstone (SRS_09), showing closed and open fractures (red). Dashed box shows the location of panels (b-e). Solid black box outlines the location of panels (f-h). Grey box shows the location of panels (i-k). (b) Reflected light photomicrograph showing substantial grain crushing (top of frame) at the crater floor and a high number of trans- and intergranular fractures in the region beneath. Interconnected fracture pathways are seen in the fracture map in panel (c). The highest fracture intensity value (0.124) is observed in the lower right of the $P_{21}$ intensity map (d), 5.9 mm from the crater centre (out of frame towards the top right). Topology parameters were calculated using the branch network (black lines) interpreted by the NetworkGT plugin based on a threshold image of the digitised fractures (red lines). The orientations of the digitised fracture network show a slight predominance in orientation around 45°/225° and 90°/270°. (f-g)

Reflected light micrograph and corresponding fracture map of open, extensional fractures directly below the crater edges. Fractures are oriented sub-parallel to the target face as seen in the rose diagram for the region (h). (i) Reflected light photo micrograph and fracture network showing a region of fracturing from below the crater to the SW corner of the central section. (j) $P_{21}$ fracture intensities and NetworkGT branch map for the same region. There appears to be a slight trend of fracture orientations from 45°/225° (k), though the dominant orientation for the region is perpendicular to the target face. All rose diagrams (e, h, k) are plotted as equal area diagrams, orientation frequency is weighted for fracture length, and the radial scale is the square root of the weighted frequency.

dipping outer spall zone surrounding a deeper, flat-bottomed pit. The inflection point between these two regions on the crater edges forms an overhang with the upper wall of a large open fracture. The open fracture has a gently convex up shape across multiple thin sections, reaching the edge of the target block (Fig 6a). It was noted during thin section production that this fracture reaches the surface of faces adjacent to the impacted face. The exposure in thin section represents a 2D profile through an axisymmetric, roughly conical fracture plane with its apex at the impact crater. The aperture of the open fracture is widest (~13 mm) where it intersects the crater, narrowing to ~1.5–2 mm near the edge of the target block. This fracture forms a wedge of material (incipient wedge) that appears to be unconnected to the rest of the target block within the plane of observation. Peak $P_{21}$ values in the limestone target are lower than those in sandstone (0.053 vs. 0.124), with high $P_{21}$ values localised in the near surface region of the spall zone in the top right section, beneath the crater floor, and around the open fractures

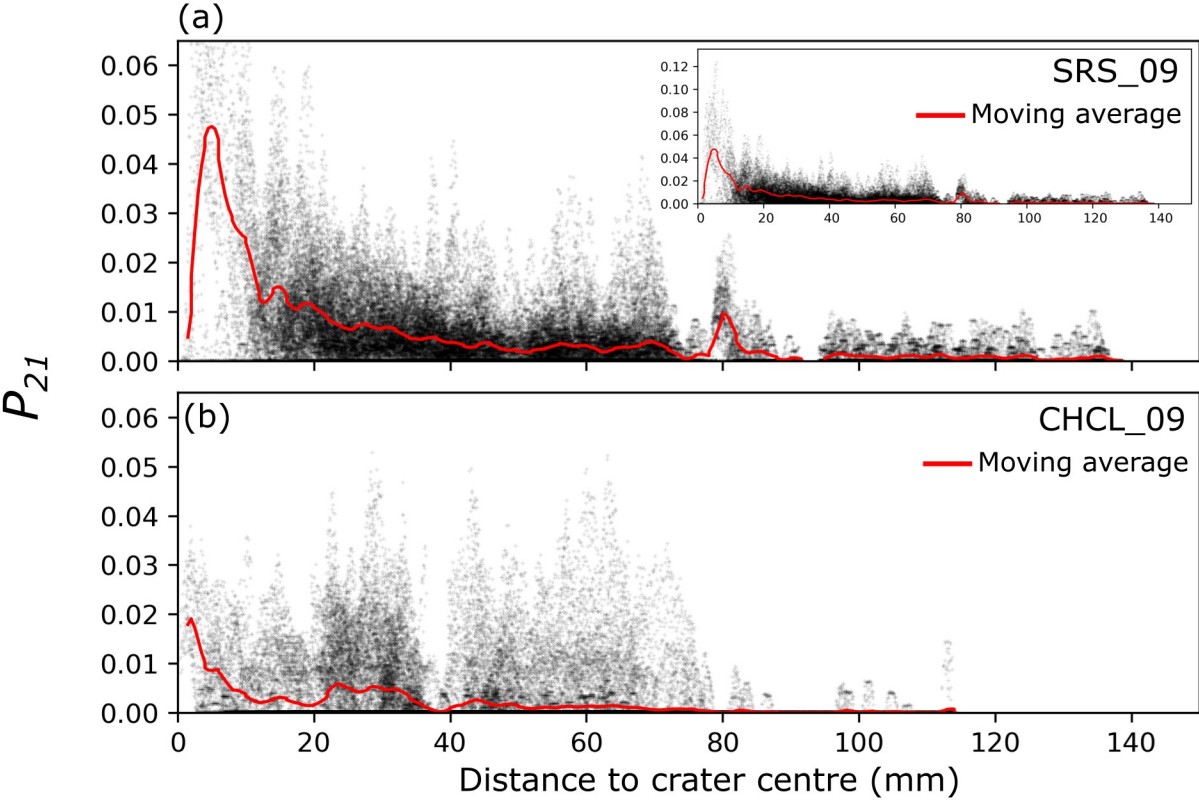

**Fig 5. $P_{21}$ fracture intensity with increasing distance from the crater centre for the sandstone (a) and limestone (b) target blocks.** Red line is a 2 mm moving average of $P_{21}$ intensity with distance from crater centre. Inset shows the full extent of $P_{21}$ values in the sandstone target.

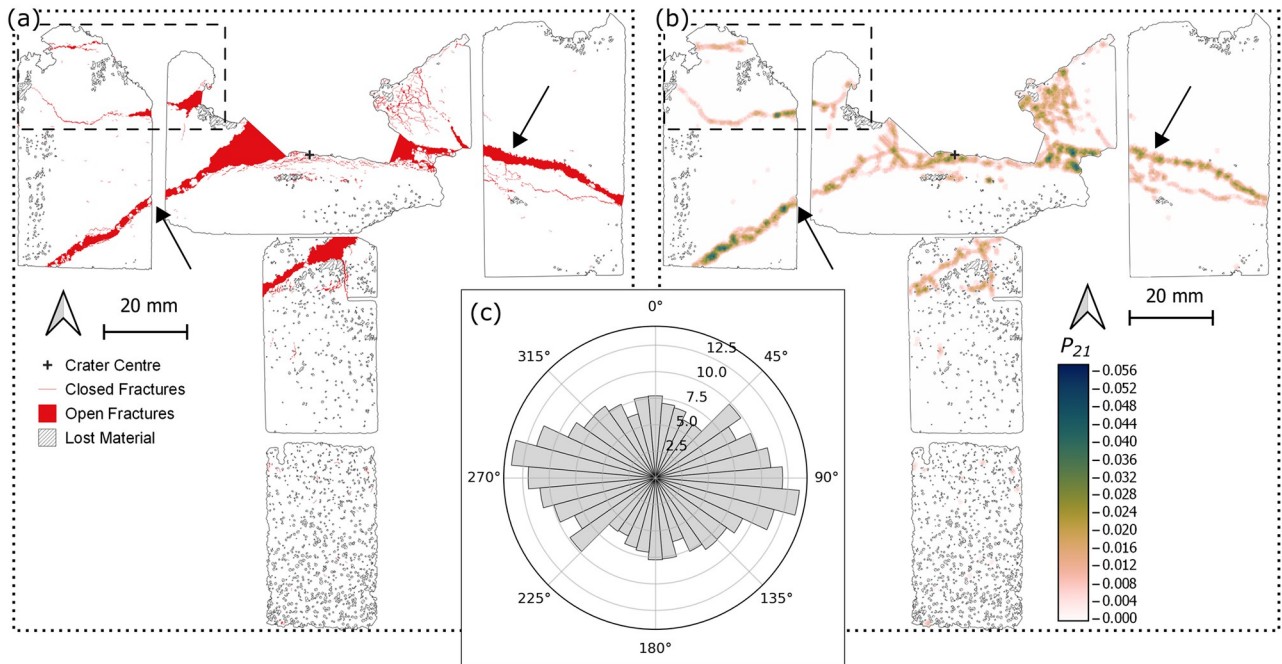

**Fig 6.** Fracture map (a) through the centre plane (*XZ*) of the Cotswold Hill Cream Limestone (CHCL) sample (CHCL_09). An open fracture (black arrows) is present across multiple thin sections, intersecting the edges of the target block and the crater. Open fractures are visible sub-parallel to the target face and forming incipient spall fragments (dashed rectangle). There are crater floor parallel, closed fractures (red line) directly below the crater centre. (b) Map of $P_{21}$ fracture intensity values across the thin sections. The highest values (dark blue) are localised along the wide open fracture (black arrows) and around the crater centre. For both maps impact direction is top to bottom and the original block outline is shown with a dotted line. (c) Equal area rose diagram showing the orientation of all fractures, weighted for fracture length, mapped within the limestone sample. The fractures are predominantly sub-parallel to the target face. Radial scale is the square root of frequency.

(Fig 6b). The highest $P_{21}$ intensity values are within ~5 mm from the crater centre, decreasing by at least a factor of 2 beyond this distance (Fig 5b).

Fractures throughout the sample are generally sub-parallel to the target face (Fig 6c), although there is another group of fractures with an orientation of 50˚/230˚. Material below the spall zone surface is highly fractured, with grain sizes beyond the scale of observation in optical sections (Fig 7a–7c). The top surface of the incipient wedge is the floor of the spall zone surrounding the central excavation and has an orientation of approximately 45˚/225˚. Some fractures within the wedge, particularly those close to the spall surface, are oriented parallel to the spall surface, while other fractures throughout the wedge are perpendicular to this surface (Fig 7c). Higher $P_{21}$ values reflect the higher fracture intensity in these regions (Fig 7d). This orthogonal pair of fractures is bisected by a third group, with orientations of approximately 100˚/280˚ (Fig 7e.)

Clasts of wall rock occur within the large open fracture that crosses multiple thin sections. There are several narrower (< 0.15 mm) open fractures sub-parallel to, but distinct from, the large fracture (Fig 7f). Up to 2 mm beneath the floor of the central excavation there is a set of open fractures <0.2 mm wide and parallel to the crater floor. 6 mm below the crater floor is a zone of crushed ooids and very fine grained material, below the scale of observation (Fig 7g). There is another large open fracture (0.6–5.5 mm wide) starting at least 20 mm below the crater floor and oriented towards the lower left of the block (in section view), intersecting the edge of the section area at a depth of 30 mm below the crater floor (thin section below the crater centre in Fig 7a).

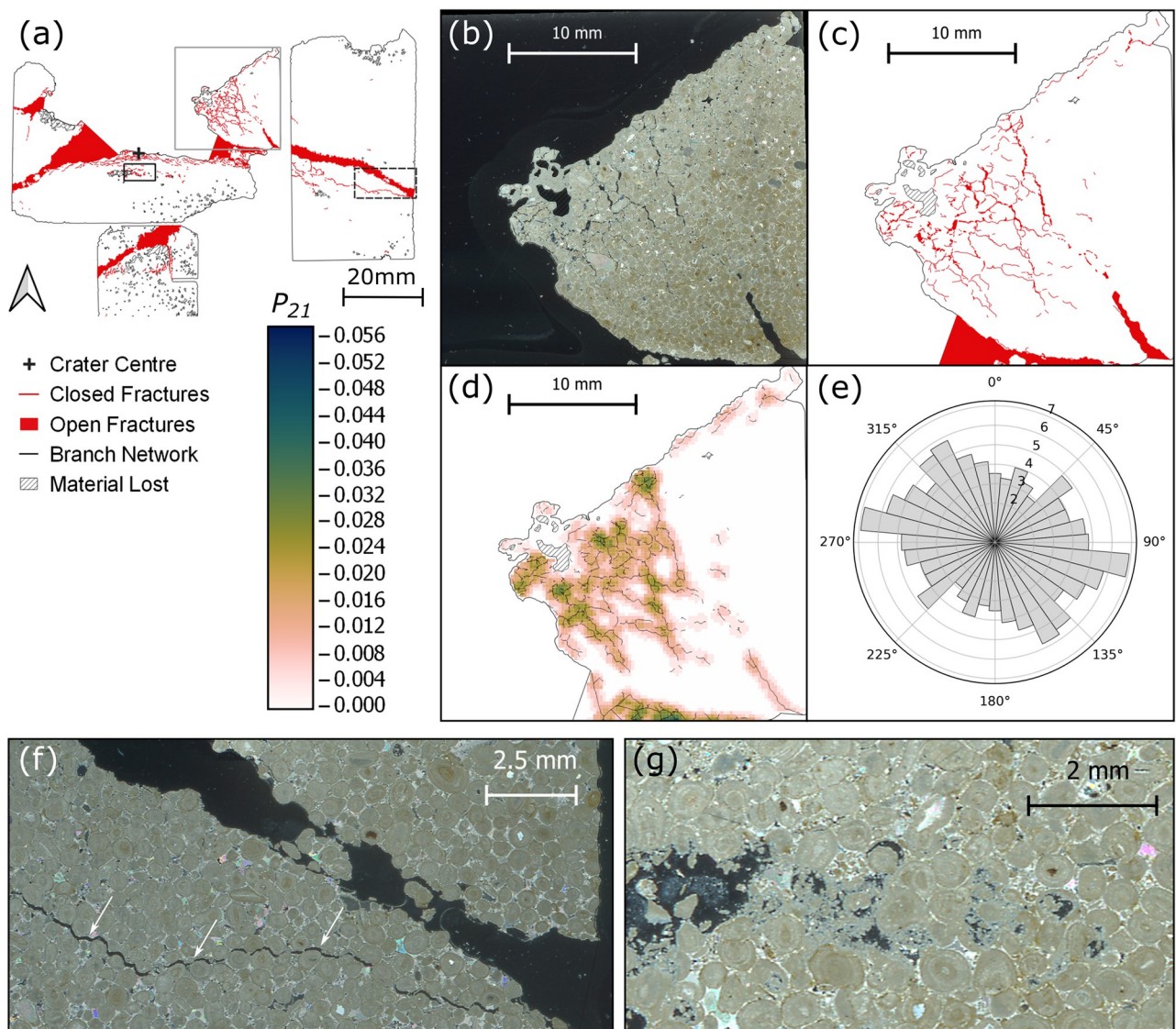

**Fig 7.** (a) Fracture map of the top central and right thin sections of sample CHCL_09 showing closed (red line) and open (solid red) fractures. Grey box indicates the location of panels (b-e), dashed black box indicates location of panel f and solid black box panel g. (b) Photomicrograph taken under cross polarised light (XPL) of an incipient wedge formed at the edge of the crater. (c) Fracture map showing multiple orientations of open and closed fractures in the wedge, corresponding to increased $P_{21}$ intensity, as shown in panel d. (e) Equal area rose diagram of length weighted fracture orientations in panels b-d. Radial scale is the square root of frequency. (f) Photomicrograph under cross polarised light of a large open fracture present across several sections that intersects the edge of the target block. The fracture contains clasts of wall rock and has narrower fractures sub-parallel to it but several mm away (white arrows). (g) Photomicrograph under XPL highlighting a region of crushed ooids and carbonate material 6 mm below the crater floor.

## Discussion

### Damage mechanics

The experiments conducted here were carried out at conditions intermediate between hypervelocity and quasi-static experiments (Table 2), with potentially some overlap between the conditions for these ordnance impacts and those of hypervelocity impacts. Strain rates of $10^3-10^6$ s$^{-1}$ here compare with $10^4-10^9$ s$^{-1}$ for hypervelocity experiments and $<10^3$ s$^{-1}$ for

**Table 2. Summary of the similarities and differences in damage appearance and mechanisms for hypervelocity impacts, ordnance velocity impacts, and quasi-static indentation experiments.**

| | Hypervelocity Impact | Ordnance Velocity Impact (This Study) | Quasi-Static Indentation |
|---|---|---|---|
| Strain Rate (s$^{-1}$) | $10^4$–$10^9$ | $10^3$–$10^6$ | $<10^3$ |
| Impact velocity (ms$^{-1}$) / P-wave Velocity (ms$^{-1}$) | 0.9–2.9 | 0.66–0.94 | ~$10^{10}$ |
| Spall fractures | ✓ | ✓ | - |
| Conical fractures or zones of fracture | At the boundary of the near surface zone | 5–10× the depth of the near surface zone | Cone cracks |
| Radial Fractures | ✓ | ✓ | ✓ |
| Concentric fractures | ✓ | ✓ | ✓ |
| Crater Mechanics | A point source equivalent to an explosion at depth | Momentum transfer | Quasi-static crack growth |
| References | [25–27] | [28–30] | [27, 31–33] |

quasi-static experiments. Another way to compare the experimental conditions is the ratio of impact velocity to P-wave velocity in the target: these experiments have values of 0.66 to 0.94 compared to the values of 0.9 to 2.9 for hypervelocity and ~$10^{10}$ for quasi-static experiments. Despite these considerable differences, there are several features in common between the different experiments (Table 2).

The open fracture observed in the limestone sample dipping away from the crater resembles the 'near surface' fractures observed below hypervelocity impacts into gabbro (Fig 8) [34]. Polanskey and Ahrens [34] suggest that the fractures form along the boundary between a near surface region, as defined by Melosh [35], and deeper regions of the target. In the near surface region, target material experiences reduced peak compressive stress due to the reflection at a free surface of compressive stress waves as tensile waves of equal magnitude. As rock is generally weaker in tension than compression, these tensile waves can overcome rock strength and result in extensional fracturing, i.e. spallation. Polanskey and Ahrens [34] show good correlation of both location and orientation between the boundary of the near surface zone and 'near surface' fractures below hypervelocity impacts. Calculation of the near surface boundary for the experiments conducted here, as defined in Melosh [35] (Eq 2), resulted in a depth below target surface ($Z_p$) of 4.2–9.3 mm for the limestone experiment (Fig 9a) and 4.1–14.2 mm for the sandstone experiment (Fig 9b).

$$Z_p = \frac{C_L T}{2} \left( \frac{4(d^2 + s^2)}{4d^2 - C_L^2 T^2} \right)^{\frac{1}{2}} \tag{2}$$

Where $C_L$ is the target sound speed, $T$ is the rise time of the stress pulse (and $T \approx a/U$) where $a$ is projectile diameter and $U$ is its impact velocity, $d$ is the depth of burst, and $s$ is the distance along the surface ($X$ axis) from the impact point. The depth of burst is the effective centre of the spherical stress wave that diverges from the impact site and defined here as $d \approx 2a(\rho_p / \rho_t)^{1/2}$ with $\rho_p$ the projectile density and $\rho_t$ the target density [35]. In this equation $d$ is similar to, but not the same as the 'depth of burst' for an explosion that produces a crater the same size as the impact, a common reference depth used in hypervelocity experiments. The value of $d$ (38.8 mm) for the limestone target is similar to the maximum crater depth (42.5 mm), a similarity not observed in the sandstone target (36.6 vs. 5.1 mm). For both targets in this study, the theoretical hyperbola of the near surface boundary does not have a strong

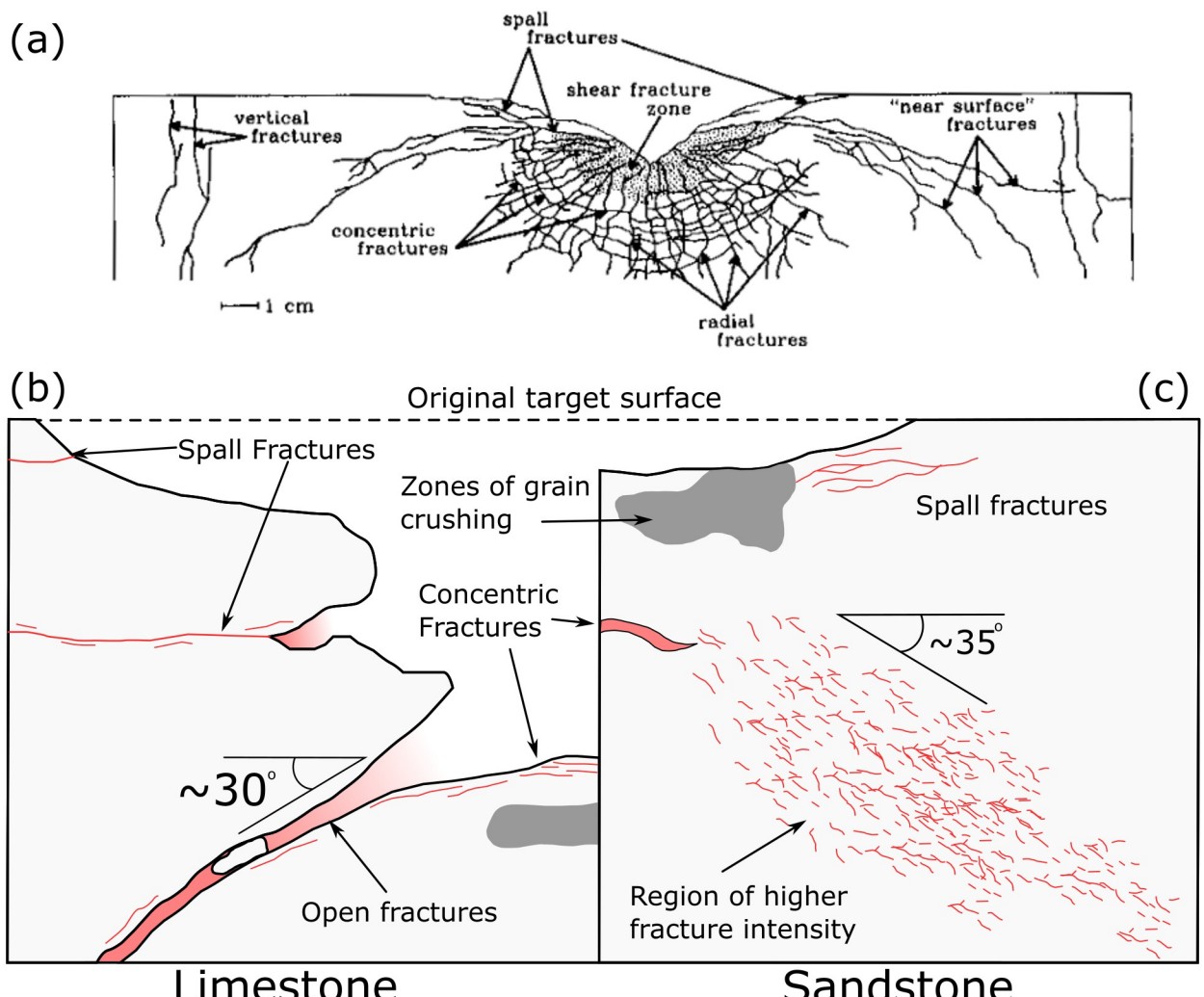

**Fig 8.** (a) Summary sketch of damage to San Marcos Gabbro during a hypervelocity impact [34]. Schematics (not to scale) of damage observed in limestone (b) and sandstone (c) targets shot with 7.62 x 39 mm ammunition.

correlation with the observed subsurface fracturing (Fig 9). Fractures are present in the near surface zone of the sandstone target, but they are parallel to the crater floor or target surface, comparable to those labelled 'spall fractures' by Polanskey and Ahrens [34] (Fig 8a). One experiment of Polanskey and Ahrens [34], using a commercial lead bullet fired at 890 ms$^{-1}$, created near surface fractures with a steeper inclination than predicted by their theoretical near surface parabolas. The results of this experiment resemble the orientation of the increased fracture intensity zone in the sandstone target of this study. Winkler et al. [36] observed localised shear zones below hypervelocity impacts into quartzites that dip away radially from the crater centre, some of which have orientations similar to those observed in the sandstone target of this study. The shape of the near surface zone is strongly controlled by the stress pulse caused by the impact [34, 35]. The model discussed above assumes the rise time remains constant as shock/stress propagates [35], which is unlikely for the ogive nose shape of the projectile in this study.

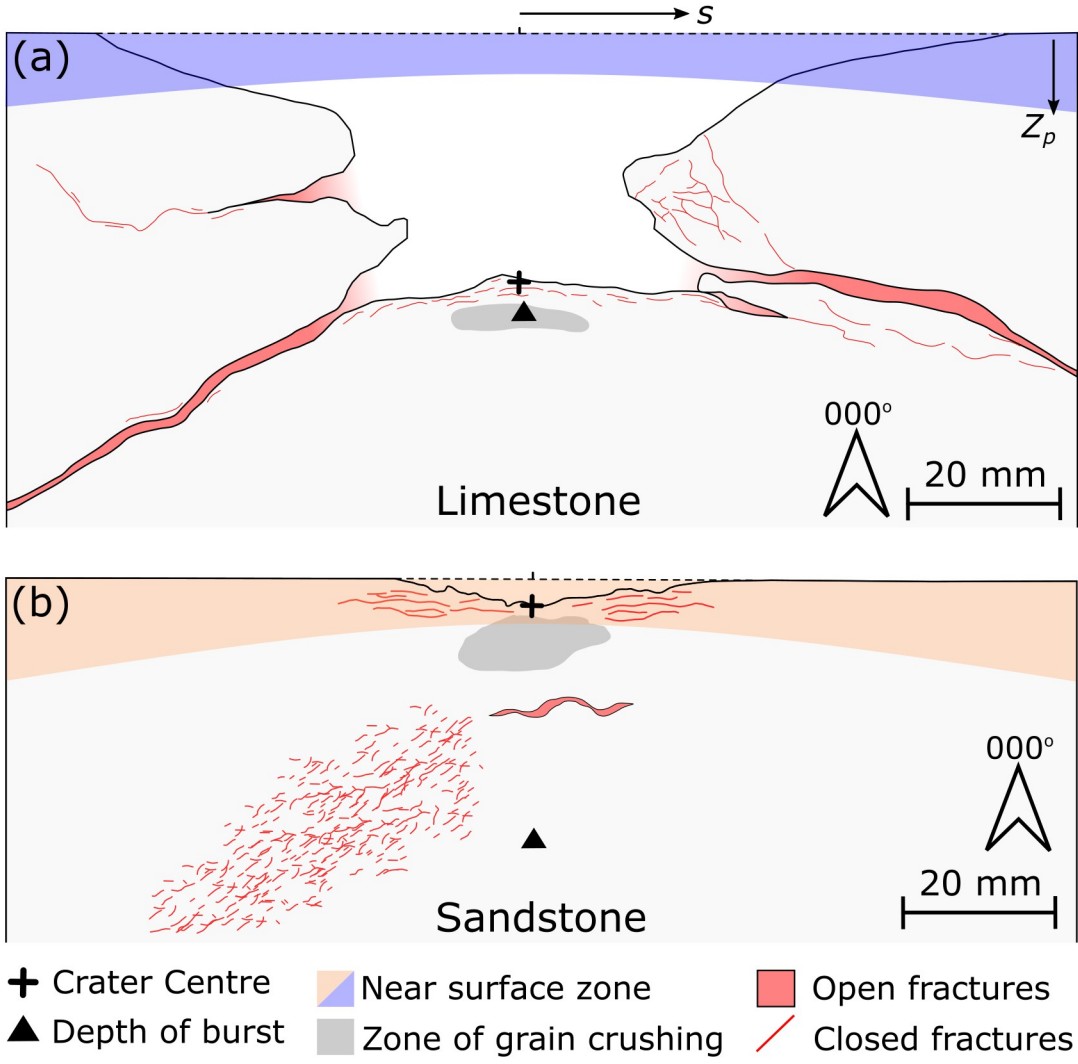

**Fig 9.** (a) Summary diagram of damage to Cotswold Hill Cream Limestone. The predicted depth of burst (d) (triangle) and crater centre are a similar distance below the original target face (dashed line). $Z_p$ is the depth of the near surface zone parabola at lateral distance (s) from the impact point. The theoretical near surface zone is shaded blue. (b) Summary diagram of damage to Stoneraise Red Sandstone. The predicted depth of burst (d) (triangle) is substantially deeper in the target than the crater centre. $Z_p$ is the depth of the near surface zone parabola at lateral distance (s) from the impact point. The theoretical near surface zone is shaded orange. Vertical and horizontal scales are the same.

The conical form of the subsurface fractures in the target lithologies presented here also resemble conical cracks below indentation and contact loading studies into glass and ceramic targets [37–41]. Cone fractures, also known as Hertzian cracks, form initially as a ring crack around an indentor, before propagating in a conical form with continued load. It is conventionally assumed that the orientation of the cone crack matches the pre-existing stress field, making an angle of approximately 30˚ to the surface [42], which is similar to the angle of the fracture in the limestone target and the zone of increased fracture intensity in the sandstone target. Cone cracks are considered to propagate stably, requiring quasi-static conditions [43–46]. However, impact induced fracturing is generally thought to be a dynamic process, leading to multiple flaws propagating unstably instead of a single, stable fracture [25, 47, 48].

Furthermore, the cone crack experiments use target materials with no porosity, contrasting with the relatively porous (11–20%) targets presented here. Chen et al. [2016] observed radial fractures around an indentor for target porosities between 5% and 45%, but no Hertzian cone cracks. They suggest this was due to the small radius of the indentor and relatively low target hardness resulting in plastic deformation before the critical load for cone crack formation could be reached. Impacts of a flat ended projectile into granite tiles at velocities of 207–537 ms⁻¹ by Hogan et al. [49] created conical cracks that reached the rear face of the target tiles. Other experiments impacting spherical projectiles into fused-silica and Pyrex targets, at velocities up to 340 ms⁻¹, also resulted in conical cracks below the impact [41]. Similar impacts in the same study, but into soda-lime glass targets, produced an array of splinter cracks that resemble dynamic fracturing more than stable propagation, suggesting that target material has an influence on cone crack formation from impacts [41]. The loading rate (25 μms⁻¹) of Chen et al.'s [50] indentation experiments is orders of magnitude slower than experienced by the experiments of Chaudhri [41], Hogan et al. [49], and those presented here. Both Chaudri [41] and Hogan et al. [49] described these conical fractures as Hertzian cone cracks, but their similarity to the experiments here, the limestone target in particular, suggests an alternative dynamic mechanism.

The propagation of radial fractures is observed in hypervelocity, ordnance velocity, and quasi-static indentation experiments. Radial fractures form due to tensile stresses perpendicular to the spherical compressive stress (or shock) wave caused by contact loading or impact into a target [51]. Chen et al. [50] observed four radial fractures in glass targets at orthogonal orientations around the indentor. They suggest the propagation of fractures in these orientations relieves stress in the interim regions, meaning that the growth of the four fractures accommodates the increasing indentation load. The radial fractures observed in hyper- and ordnance velocity experiments are more numerous and have less regularity in their spacing. Impact loading creates far greater strain rates (Table 2) compared to those in Chen et al.'s [50] experiments, possibly exceeding the ability of only a few orthogonally oriented radial fractures to accommodate strain, resulting in new fractures forming in the intervening areas. The propagation of multiple fracture strands at once is indicative of dynamic fracturing, observed by Hogan et al [49] and Chaudhri [41].

Both target lithologies of this study exhibit extensional fractures parallel to the crater floor, resembling observations of concentric fractures below hypervelocity impacts [26, 34, 50, 52] (Fig 8a and 8c). Similarly concentric fractures are also present beneath point loading experiments in glass and ceramics. However the fractures beneath the point loading experiments are thought to be caused during the unloading phase, as the load on the compressive zone below the indentor is released [50, 53].

Both hypervelocity and ordnance velocity impacts exhibit spall fractures at the edge of the crater. Where not directly visible in the subsurface, the presence of spallation is evident in the shallow dipping region surrounding the central excavation [24, 26, 34]. The spall fractures form when the initial compressive stress wave reaches the free surface of the target face and reflects as a tensile wave of equal magnitude [35]. Spall fractures are typically found close to the target face because the radial decay function causes wave energy to drop below the failure strength of the target material [34, 35, 54]. There are no spall fractures in quasi-static indentation experiments because the loading rates do not produce a stress wave of substantial magnitude. Instead the continual loading increases compressive stresses in the region directly below the loading.

The observations in this study have some similarities to those in both the near surface zone of hypervelocity experiments and Hertzian cone cracks, but different mechanisms involved in these ordnance velocity impacts preclude either the hypervelocity or cone crack mechanics

from fully explaining the observations made here. The formation of spall fractures parallel to the target face and crater floor show that a tensile stress wave formed when the initial compressive stress wave was reflected at the surface. The interaction of these waves reflecting from the impacted face and adjacent sides of the target block may have caused regions of tensile failure, similar to the formation of the near surface zone in the hypervelocity experiments. However, the mechanics of the ordnance impacts, involving momentum transfer and longer interaction time between the projectile and target, and the geometry of the target blocks has resulted in a sufficiently different expression of subsurface damage that the theoretical near surface zone is not applicable. The hypervelocity ($>$1500ms$^{-1}$) experiments used spherical projectiles and cratering in these experiments was primarily controlled by the generation of a shock wave originating at some depth below the surface, but these conditions and processes may not be applicable to experiments presented here. Campbell et al. [16] found that bullet impacts with velocities of 400–900 ms$^{-1}$ did not follow crater scaling relationships found in hypervelocity impacts. They also found that impact craters had identifiable crater asymmetry when impact trajectories were oblique [24]. This asymmetry is not observed in hypervelocity impacts, except for those with very oblique trajectories ($<$15˚ to target face), because of the symmetrical nature of the point source model for hypervelocity cratering mechanics. Campbell et al. [24] suggest they observed crater asymmetry in their experiments because the impact velocity was lower than, or similar to, the sound speed of the target materials, so no shockwave was generated upon impact. Cratering was instead controlled by momentum transfer from the projectile to the target. This invalidates the application of the point source assumption, critical to hypervelocity, to these experiments. The impact velocities in this study (532 ms$^{-1}$ and 539 ms$^{-1}$) are lower than the respective P-wave velocity of the limestone (569 ms$^{-1}$) and sandstone (822 ms$^{-1}$) targets, so the generation of a shock wave at impact is unlikely. The results presented here support the suggestions made by Campbell et al. [16] that bullet impacts into stone are predominantly controlled by target properties, primarily material strength. Although there are some similarities between the damage created by hypervelocity experiments and this study, such as the near-surface fractures, spalling, and grain crushing below the impact, the damage mechanisms in each case are probably different.

## Implications for conservation

Fractures play a fundamental role in the transport of moisture and weathering agents by increasing porosity and linking together isolated pores within the stone [55, 56]. Both stone types analysed here have increased fracture intensity in the regions proximal to the bullet impact, as well as regions of increased fracture intensity or open fractures dipping away from the impact crater at about 30˚. Fracture width and intensity influence fracture capacity and transmissivity, with fracture intensity strongly correlated to overall permeability [56, 57]. The pattern of higher fracture intensities closer to the crater centre suggests that regions directly surrounding the impact will have the highest induced porosity and permeability, and may therefore be at the highest risk of weathering from moisture related processes. The large open fractures present in the limestone target creates localised areas of high fracture intensity that penetrate deep into the block. Higher fracture intensity has been linked to greater rates of weathering [14].

Because the open fractures dip away from the crater centre, most of the fracture is not visible from the surface. Hidden subsurface damage may therefore affect a much larger region than visible surface damage. Fractures that intersect the sides of the impacted block can break along the mortar block boundary, or the mortar itself, possibly destabilising a wider region than just the impacted block [58]. Impact craters, particularly from shrapnel, commonly do

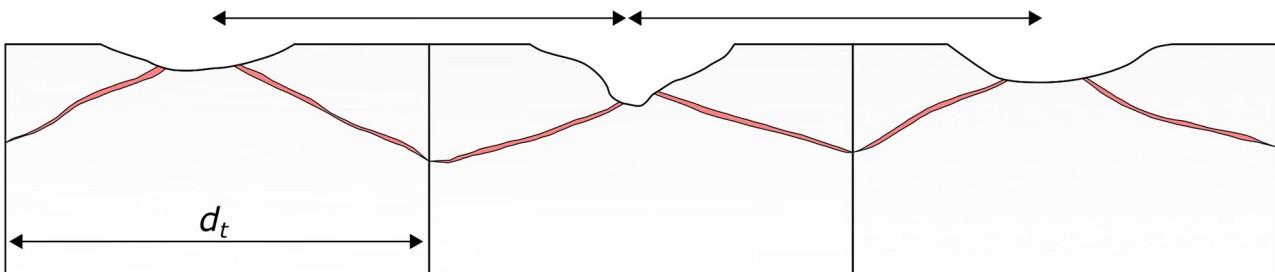

**Fig 10. For impacts with a spacing less than the diameter of the impacted block *a*, subsurface conical fracture and damage zones can form an interconnected network that affects a greater region than suggested by the surface damage alone.**

not occur in isolation; structures typically have multiple impacts across their surface. If these impacts have subsurface damage zones similar to those in this study, there is the possibility they may link up in the subsurface. Fig 10 illustrates how multiple impacts with a spacing less than the impacted block diameter may form interconnect fracture networks below the surface that have a much greater footprint than the observable surface damage. The increased permeability and decreased stone strength resulting from the interconnected damage zones may exacerbate material loss and greatly increase degree and rate of future deterioration. The interaction of subsurface damage from multiple impacts is an interesting and important avenue for future research.

The limestone target in this study has lower fracture intensities throughout, despite exhibiting greater surface damage than the sandstone sample. The $P_{21}$ values of the limestone target do not a show a sharp increase within 10mm of the crater centre, as observed in the sandstone target. Energy above the requirement to exceed the target strength can be transferred as kinetic energy, causing material to be ejected from the impact site as ejecta, or the surrounding areas as spall fragments [35]. The lower tensile strength of the limestone compared to the sandstone may explain the larger crater dimensions in the limestone target. The maximum depth of the limestone crater is 42.5mm, 8 times deeper than the crater in the sandstone. The region of highest fracture intensity in the limestone target may thus have been ejected.

The observations of impact induced fracturing in this study are important for post-conflict conservation of damaged heritage. Surface parallel spall fractures and interconnected subsurface conical fractures mean that regions with multiple impacts in close proximity may require rapid stabilisation to prevent substantial material loss. The increased permeability and porosity surrounding the impact mean these regions are at increased risk from moisture related deterioration (e.g. dissolution, salt crystallisation), so efforts for protecting against moisture, such as erecting temporary rain covers or shelters, can be prioritised where impacts are most numerous or exposed. Rapid observation of surface damage can suggest where these priority actions should be focussed for short term protection. Once a more detailed and comprehensive assessment of the damage and risk of deterioration has been undertaken, then targeted and specific remediation efforts can be conducted. It is acknowledged that heritage can be constructed from a variety of materials, and that this study only investigates two of the most common stone types used. It would be useful in particular to determine if similar damage occurs in crystalline, low porosity targets.

## Conclusion

Apart from the visible surface crater, bullet impacts into rocks create conical fractures or zones of increased fracture intensity below the impact, radial fractures, and spallation. Similar

features are also seen in hypervelocity experiments and quasistatic indentation experiments that form cone cracks. However, the strain rates and impact velocities of bullet impacts are intermediate between the hypervelocity and quasistatic experiments, and the mechanisms causing damage are distinct from these experiments. Fracturing from the bullet impacts was dynamic (unlike cone crack experiments) but a shock wave did not form (as in hypervelocity experiments). Damage was caused by momentum transfer. The distinct conditions and damage mechanics in the bullet impacts created differences in the details of the geometry of their damage compared to the faster and slower impacts.

The subsurface damage caused by bullet impacts differs between target lithologies. Sandstone exhibits predominantly closed aperture inter- and intragranular fracturing, with some open fractures sub-parallel to the target face, as well as zone of grain size reduction and compaction directly below the crater. Limestone exhibits open fractures parallel to the target surface, and open fractures dipping away from the crater at 30° and propped open by clasts of wall rock. These open fractures can intersect sides of the target adjacent to the impacted face, potentially leading to the loss of large volumes of material.

$P_{21}$ fracture intensity is highest closer to the crater centre in both lithologies and greatly decreases beyond 5–10 mm from the crater centre. This shows that the region directly surrounding the crater centre is at the greatest risk of deterioration from weathering. Regions at risk are not limited to the impact crater, open fractures and zones of higher fracture intensity adjacent to them provide conduits for moisture ingress and regions of increased susceptibility to weathering processes. These fractures have the potential to link up with subsurface fractures below adjacent impacts and exacerbate the risk of future deterioration from weathering processes across a much larger area. Small and apparently inconspicuous impact craters have subsurface damage that can extend up to 80 mm from the target face into the targeted block, but have little to no visible surface expression. This is important for proper surveying and post conflict risk assessments of heritage sites.

## Supporting information

**S1 Appendix. Description of uncertainty calculation in fracture analysis.**
(DOCX)

**S1 Data. CHCL_09 fracture analysis data.**
(ZIP)

**S2 Data. SRS_09 fracture analysis data.**
(ZIP)

## Acknowledgments

We would like to thank the technicians and staff at Cranfield Ordnance Test and Evaluation Centre (COTEC) for their expertise and assistance in conducting the live fire experiments. Additional thanks to Lieutenant Martin RE for information and discussion surrounding engagement distances. Finally we would like to thank Anthony Oldroyd for their valuable assistance during sample preparation.

## Author Contributions

**Conceptualization:** Tom Blenkinsop, Lisa Mol, Oscar Gilbert.

**Data curation:** Oliver Campbell.

**Formal analysis:** Oliver Campbell.

**Funding acquisition:** Lisa Mol.

**Investigation:** Oliver Campbell, Oscar Gilbert.

**Methodology:** Oliver Campbell, Oscar Gilbert.

**Software:** Oliver Campbell.

**Supervision:** Tom Blenkinsop, Lisa Mol.

**Visualization:** Oliver Campbell.

**Writing – original draft:** Oliver Campbell.

**Writing – review & editing:** Oliver Campbell, Tom Blenkinsop, Lisa Mol, Oscar Gilbert.

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
