## [Decision Letter · Decision Letter 0]

3 Apr 2023

PONE-D-22-33228

Subsurface fracturing of sedimentary stones caused by bullet impacts

PLOS ONE

Dear Dr. Campbell,

Thank you for submitting your manuscript to PLOS ONE. After careful consideration, we have decided that your manuscript does not meet our criteria for publication and must therefore be rejected.

The article is not correct scientifically and written like a technical report. 

I am sorry that we cannot be more positive on this occasion, but hope that you appreciate the reasons for this decision.

Kind regards,

Yasir Nawab, PhD

Academic Editor

PLOS ONE

Additional Editor Comments:

The article is not sound technically, therefore, it cannot be considered further in its current form.

Some comments for improvement are:

1. It is suggested to the author to do the extensive literature review to do research on these types of damages.

2. The author has no basis knowledge about multilayer woven fabrics and multilayer composites.

Please study the basis terminologies about the construction field.

2. This study could be a report for only one building, but as the bullet is concerned, there are numerous structures and material for building construction so this study could not be helpful to draw any conclusion for benefit at large scale.

3. This study is focused on the political issues rather than scientific problems, so it could not be accepted as a research publication. Figure 1.

4. There is no specific and valuable objective of this study.

5. The references are with incomplete details.

Reviewers' comments:

Reviewer's Responses to Questions

**Comments to the Author**

1. Is the manuscript technically sound, and do the data support the conclusions?

Reviewer #1: Yes

Reviewer #2: No

2. Has the statistical analysis been performed appropriately and rigorously? 

Reviewer #1: Yes

Reviewer #2: No

3. Have the authors made all data underlying the findings in their manuscript fully available?

Reviewer #1: Yes

Reviewer #2: No

4. Is the manuscript presented in an intelligible fashion and written in standard English?

Reviewer #1: Yes

Reviewer #2: No

5. Review Comments to the Author

Reviewer #1: The MS examines the effects of bullet damage on building stones. The paper is clearly written and well illustrated, and the topic is likely of non-trivial interest. The fracture analysis seems to be correct.

The work is primary scientific research. The experiments are appropriate and well explained. The fracture analysis is excellent. The conclusions are supported by the data and the discussion.

76 The strength of sandstones and limestones varies a lot with compaction, natural cement, and porosity. This is true even amongst building stones. To help make these results more widely transferable In addition to the mechanical properties that are provided I suggest providing a bit more information about the petrography (cements, porosity) of these two rock types. I see some porosity values in line 252. Did I miss something earlier?

82 ‘Inherent’ fractures seems cryptic. Maybe this is a term or art in building stones? Does this mean ‘no pre-existing’ fractures? Does this include naturally occurring microfractures? Some building stones have strength anisotropy due to microfracture arrays. These can be diagnosed (Anders et al. 2014, J. Str. Geol.)

105 Were any precautions like pre-sectioning application of epoxy taken to minimize or eliminate sample preparation damage? (see also 144)

112 What do you mean by ‘closed fractures’? Does this mean that no open space was visible at the magnifications used? Or are these somehow otherwise closed? Later you use the term ‘open extensional fractures’, so could ‘closed fractures’ signify a different fracture mode? Probably best to clearly define what you mean (categories) at the outset.

152 Interesting texture. Is this a very porous sandstone?

160 Since such ‘within-grain’ fractures are naturally occurring in many sandstones (Anders et al. 2014) do you have samples from distant from the impact sites to corroborate that some of these are not part of a naturally occurring population?

Fig. a, b. Although I suppose the scale is obvious, it would be a good idea to put graphic bar scales on the images. Check other diagrams for scales.

Reviewer #2: Review Report

The research article titled “Subsurface fracturing of sedimentary stones caused by bullet impacts” investigates the subsurface damage caused by bullet impacts, which is not apparent from surface inspection, in order to better understand the geometry and mechanics of this form of conflict damage to heritage.

It is suggested to the author to do the extensive literature review to do research on these types of damages.

The author has no basis knowledge about multilayer woven fabrics and multilayer composites.

Please study the basis terminologies about the construction field.

1. This study could be a report for only one building, but as the bullet is concerned, there are numerous structures and material for building construction so this study could not be helpful to draw any conclusion for benefit at large scale.

2. This study is focused on the political issues rather than scientific problems, so it could not be accepted as a research publication. Figure 1.

3. There is no specific and valuable objective of this study.

4. The references are with incomplete details.

I found no new findings in this research article.

This article has a lot of language errors, and no proper scientific presentation is followed.

This work can be report but not the novel idea or any new addition in previous practices, so it is Rejected for publication in the journal. The language and writing style is so unethical according to research norms.

6. PLOS authors have the option to publish the peer review history of their article (what does this mean?). If published, this will include your full peer review and any attached files.

Reviewer #1: No

Reviewer #2: No

- - - - -

---

## [Author Response · Author response to Decision Letter 0]

8 May 2023

Reviewer #1

The MS examines the effects of bullet damage on building stones. The paper is clearly written and well illustrated, and the topic is likely of non-trivial interest. The fracture analysis seems to be correct.

The work is primary scientific research. The experiments are appropriate and well explained. The fracture analysis is excellent. The conclusions are supported by the data and the discussion.

76 The strength of sandstones and limestones varies a lot with compaction, natural cement, and porosity. This is true even amongst building stones. To help make these results more widely transferable In addition to the mechanical properties that are provided I suggest providing a bit more information about the petrography (cements, porosity) of these two rock types. I see some porosity values in line 252. Did I miss something earlier?

- The porosity of each lithology is provided in the material descriptions (Lines 73 and 78).

- The cement type for each lithology has been added into the lithology descriptions on line 73 ‘With a porosity of 11% and intergranular cement comprising quartz overgrowths, it is generally massive, with some target blocks exhibiting visible beds of coarser grains (∼1 mm) (Figure 2a).’ and line 79 ‘The majority of intergranular cement comprises sparry calcite, though areas of smaller grain sizes tend towards microcrystalline calcite cement.’

82 ‘Inherent’ fractures seems cryptic. Maybe this is a term or art in building stones? Does this mean ‘no pre-existing’ fractures? Does this include naturally occurring microfractures? Some building stones have strength anisotropy due to microfracture arrays. These can be diagnosed (Anders et al. 2014, J. Str. Geol.)

- The term inherent has been replaced throughout with ‘pre-existing’ for clarity.

105 Were any precautions like pre-sectioning application of epoxy taken to minimize or eliminate sample preparation damage? (see also 144) 

- Samples were impregnated with epoxy resin prior to sectioning in order to minimise additional damage. Explanation of this has been added in line 108 ‘The target block was impregnated with epoxy resin prior to sectioning in order to minimise further damage to the samples.’ 

- Further mention of this has been added to line 119 ‘Some thin sections were subject to material loss during section production, though every effort was made to prevent this with the pre-sectioning application of epoxy resin.’

112 What do you mean by ‘closed fractures’? Does this mean that no open space was visible at the magnifications used? Or are these somehow otherwise closed? Later you use the term ‘open extensional fractures’, so could ‘closed fractures’ signify a different fracture mode? Probably best to clearly define what you mean (categories) at the outset. 

- The term closed fractures is defined on line 117.

152 Interesting texture. Is this a very porous sandstone? 

- The sandstone is not particularly porous, with an average porosity of @11%.

160 Since such ‘within-grain’ fractures are naturally occurring in many sandstones (Anders et al. 2014) do you have samples from distant from the impact sites to corroborate that some of these are not part of a naturally occurring population? 

- Samples from undamaged blocks of the same lithology were used to corroborate that these fractures are impact induced. This is explained on Line 83.

Fig. a, b. Although I suppose the scale is obvious, it would be a good idea to put graphic bar scales on the images. Check other diagrams for scales. 

- The images used in Figure 1a,b were not taken by the authors and no information with regards to scale is available from the source of the images.

- All other figures contain scale bars where appropriate.

Reviewer #2

Reviewer #2: Review Report

The research article titled “Subsurface fracturing of sedimentary stones caused by bullet impacts” investigates the subsurface damage caused by bullet impacts, which is not apparent from surface inspection, in order to better understand the geometry and mechanics of this form of conflict damage to heritage.

It is suggested to the author to do the extensive literature review to do research on these types of damages.

The author has no basis knowledge about multilayer woven fabrics and multilayer composites.

Please study the basis terminologies about the construction field.

1. This study could be a report for only one building, but as the bullet is concerned, there are numerous structures and material for building construction so this study could not be helpful to draw any conclusion for benefit at large scale.

2. This study is focused on the political issues rather than scientific problems, so it could not be accepted as a research publication. Figure 1.

3. There is no specific and valuable objective of this study.

4. The references are with incomplete details.

I found no new findings in this research article.

This article has a lot of language errors, and no proper scientific presentation is followed.

This work can be report but not the novel idea or any new addition in previous practices, so it is Rejected for publication in the journal. 

The language and writing style is so unethical according to research norms.

The attached review by Reviewer #2 was for a paper titled ‘Multilayer Bio composites of PLA Woven Fabric and PBS Sheets Using Compression Molding Process’.

As such, no response can be provided to the comments of Reviewer #2.

---

## [Decision Letter · Decision Letter 1]

20 Jul 2023

PONE-D-22-33228R1Subsurface fracturing of sedimentary stones caused by bullet impactsPLOS ONE

Dear Dr. Campbell,

Thank you for submitting your manuscript to PLOS ONE. After careful consideration, we feel that it has merit but does not fully meet PLOS ONE’s publication criteria as it currently stands. Therefore, we invite you to submit a revised version of the manuscript that addresses the points raised during the review process.

ACADEMIC EDITOR: Please insert comments here and delete this placeholder text when finished. Be sure to:Indicate which changes you require for acceptance versus which changes you recommendAddress any conflicts between the reviews so that it's clear which advice the authors should followProvide specific feedback from your evaluation of the manuscriptPlease ensure that your decision is justified on PLOS ONE’s publication criteria and not, for example, on novelty or perceived impact.

We look forward to receiving your revised manuscript.

Kind regards,

Anwar Khitab

Academic Editor

PLOS ONE

Journal Requirements:

“This research was funded by the Leverhulme Trust, grant number RPG-2017-408. The APC was funded by Cardiff University’s Institutional Open Access Fund.”

“The authors have declared no competing interests.”

5. We note that [Figures 3,4,6 and 7 ] in your submission contain [map/satellite] images which may be copyrighted. All PLOS content is published under the Creative Commons Attribution License (CC BY 4.0), which means that the manuscript, images, and Supporting Information files will be freely available online, and any third party is permitted to access, download, copy, distribute, and use these materials in any way, even commercially, with proper attribution. For these reasons, we cannot publish previously copyrighted maps or satellite images created using proprietary data, such as Google software (Google Maps, Street View, and Earth). For more information, see our copyright guidelines: http://journals.plos.org/plosone/s/licenses-and-copyright.

a. You may seek permission from the original copyright holder of Figures 3,4,6 and 7 to publish the content specifically under the CC BY 4.0 license.

Natural Earth (public domain): http://www.naturalearthdata.com/.

6. Please remove your figures from within your manuscript file, leaving only the individual TIFF/EPS image files, uploaded separately. These will be automatically included in the reviewers’ PDF.

Additional Editor Comments (if provided):

Reviewers' comments:

Reviewer's Responses to Questions

**Comments to the Author**

1. If the authors have adequately addressed your comments raised in a previous round of review and you feel that this manuscript is now acceptable for publication, you may indicate that here to bypass the “Comments to the Author” section, enter your conflict of interest statement in the “Confidential to Editor” section, and submit your "Accept" recommendation.

Reviewer #3: All comments have been addressed

Reviewer #4: All comments have been addressed

2. Is the manuscript technically sound, and do the data support the conclusions?

Reviewer #3: Yes

Reviewer #4: Partly

3. Has the statistical analysis been performed appropriately and rigorously? 

Reviewer #3: N/A

Reviewer #4: Yes

4. Have the authors made all data underlying the findings in their manuscript fully available?

Reviewer #3: Yes

Reviewer #4: Yes

5. Is the manuscript presented in an intelligible fashion and written in standard English?

Reviewer #3: Yes

Reviewer #4: Yes

6. Review Comments to the Author

Reviewer #3: This revised manuscript investigated the bullet impacts induced subsurface damage of two sedimentary rocks of sandstone and limestone through controlled firearm experiments. The crater, the radial fractures and spallation and the subsurface damage below the impacts were observed. The paper is clearly written and well illustrated. This manuscript has some novelty on the investigation of heritage building materials and the research is meaningful for heritage conservation. My particular comments are as follows.

1.Initial damage evaluation: Basic physical and mechanical properties of these two rocks are given but the initial damage evaluation before the impact is not given. This is important to the further damage and crack formation.

2.The effects of impact velocity and mass: A high-speed impact induced damage heavily depends the physical and mechanical properties of the rock, it also depends on the impact mass and impact velocity. These effects should be discussed.

3.This is a scientific research on the bullet impact induced damage evaluation. It is not necessary to introduce some non-scientific issues or statement. On this sense, Lines 29-35 in the Introduction are not appropriate and suitable.

Reviewer #4: I have added suggested comments in the word file. The author should incorporate these suggested comments before final acceptance.

7. PLOS authors have the option to publish the peer review history of their article (what does this mean?). If published, this will include your full peer review and any attached files.

Reviewer #3: No

Reviewer #4: No

---

## [Author Response · Author response to Decision Letter 1]

5 Aug 2023

Reviewer 3:

This revised manuscript investigated the bullet impacts induced subsurface damage of two sedimentary rocks of sandstone and limestone through controlled firearm experiments. The crater, the radial fractures and spallation and the subsurface damage below the impacts were observed. The paper is clearly written and well illustrated. This manuscript has some novelty on the investigation of heritage building materials and the research is meaningful for heritage conservation. My particular comments are as follows.

1. Initial damage evaluation: Basic physical and mechanical properties of these two rocks are given but the initial damage evaluation before the impact is not given. This is important to the further damage and crack formation.

- The pre-impact state of the rock has been evaluated by examining samples that were not subjected to bullet impact (Lines 83-85). These revealed there was no initial damage. This point is elaborated in the reply to review #4, point 8 below.

2. The effects of impact velocity and mass: A high-speed impact induced damage heavily depends the physical and mechanical properties of the rock, it also depends on the impact mass and impact velocity. These effects should be discussed.

- Table 2 and the discussion from lines 215 to 338 deal with the effect of impact velocity. As outlined in point 7 above, the number of impact variables was minimised: there is lot of new material to report from these two experiments alone.

3. This is a scientific research on the bullet impact induced damage evaluation. It is not necessary to introduce some non-scientific issues or statement. On this sense, Lines 29-35 in the Introduction are not appropriate and suitable.

- We suggest that these lines are appropriate because they are the motivation for paper and provide a compelling contemporary context; however, if the editor requests, we can remove them.

Reviewer 4:

The research paper investigates the impact of bullet damage on building stones. It is evident that the paper is well-written, with clear explanations and effective illustrations. The topic itself is likely to generate significant interest. The fracture analysis conducted in the study appears to be accurate and well-executed.

This work represents original scientific research, and the experiments conducted are appropriate and thoroughly described. The fracture analysis conducted in the study is highly commendable. The conclusions drawn from the data and the subsequent discussion are well-supported and align with the findings.

1. Abstract: Line 10- Although impacts from small arms and shrapnel leave relatively inconspicuous impact scars, they may elevate the risk of future stone deterioration. How it effects the stone deterioration any literature work (may word is used means not clear). 

- The word ‘may’ has been removed from this sentence in the abstract to avoid unnecessary ambiguity. Paragraph 2 and 3 of the introduction discuss literature on bullet and shrapnel impacts into stone, and how damage can influence future deterioration

2. Microstructural analysis of subsurface damage using thin section: It is suggested to evaluate subsurface damage using CT-Scan (or other nondestructive methods), which is a better approach than the thin section because in thin, the actual subsurface properties will be disturbed during sample preparation. 

- The authors took every precaution during the preparation of thin sections to minimise alterations and disturbance of subsurface features. This is addressed in Line 108 ‘The target block was impregnated with epoxy resin in order to minimise further damage to the sample’

- The authors are aware of the use of CT scanning for subsurface observations. We conducted test scans of the target block but found that the size of sample blocks precluded the use of such methods to analyse fractures at the same scale as thin sections. Target blocks would have needed to be subsampled to obtain the suitable resolution, introducing the same risk of additional damage as thin sections.

3. The literature part is very small and needs to extend this part. 

- The reason for the small literature part is that research on this topic is in fact very limited (which is one reason why this study is significant). The 13 references cited are the most important previous literature. We have emphasized this by changing lines 36-37 to:

‘There are very few detailed studies of the surface damage caused by bullet and shrapnel impacts, and its relationships to the subsurface.’

4. The literature about stone deterioration is not added to the literature part.

- Lines 52-61 in the literature part specifically address deterioration, and the 7 references cited are the critical ones for the study. If the reviewer has any other specific suggestions, we would be glad to consider them.

5. Some results part is discussed in the materials and method section, which should be moved to the results and discussion section (for example, line NO. 83-85). 

- The interpretation has been removed from this line, which now reads ‘Thin section micrographs from undamaged samples of each lithology show no pre-existing fractures (Figure 2). We can therefore be confident that all damage reported is the result of bullet impacts and not inherited.’

6. How many sandstone and limestone samples are tested? Not mentioned in the materials and methods section.

- Clarification of the number of samples used for the fracture analysis added to line 107: ‘Polished thin sections were cut from one damaged sample of SRS and one damaged sample of CHCL…’

7. It is better to evaluate the bullet impact from different angles and distances.

- The purpose of this study was to investigate and compare the subsurface damage between different stone types as a baseline for future comparisons. In order to do so, the number of impact variables (e.g. lithology, angle, distance) were minimised. 

- Previous investigations published by the authors [16, 24] evaluated the effects of the angle of impact. They also suggest that target material has a considerable control on the resulting damage, a conclusion supported by the results of this study.

- The effects of engagement distance are the subject of ongoing investigation.

8. In figure 2 the author stated that there was no pre-existing fracturing in the undamaged section. First the author did not show the map as shown for fractures in Figure 4. Where one can compare the results. Furthermore, the author did not take micrographs on the same scale.

- The reviewer may not appreciate that it is impossible to use an invasive technique such as thin section analysis to examine the undamaged sample. Nevertheless, following the response to point 5 above, it is clear that the damage was created by the impacts.

- Micrographs were all taken at the same scale.

9. The unit of P-wave velocity is not added.

- The unit of P-wave velocity has been added to the header in Table 2. The values in Line 208 are the ratio of two velocities and are unit-less.

10. It is suggested to add results about the mineralogy of these sandstone and limestone samples. Did mineralogy impact the fracture propagation or type of fracture?

- The mineralogy of the samples is described in lines 70-88. Extra detail has been added. Mineralogy did impact fracture propagation. This is described in the results section which contrast the sandstone with the limestone targets, and the discussion deals with the mechanical reasons for the different responses in lines 226-259. The impacts of the mineralogy have been summarized in Figure 9.

11. In Table 2 the units are not added. 

- The units for strain rate are in the column header (s-1)

- The ratio of impact velocity to P-wave velocity is unit-less. The unit of each has been added to the column header for clarity.

---

## [Editor Report · Decision Letter 2]

19 Sep 2023

Subsurface fracturing of sedimentary stones caused by bullet impacts

PONE-D-22-33228R2

Dear Dr. Campbell,

We’re pleased to inform you that your manuscript has been judged scientifically suitable for publication and will be formally accepted for publication once it meets all outstanding technical requirements.

Kind regards,

Anwar Khitab

Academic Editor

PLOS ONE
---

## [Editor Report · Acceptance letter]

28 Sep 2023

PONE-D-22-33228R2 

Subsurface fracturing of sedimentary stones caused by bullet impacts 

Dear Dr. Campbell:

I'm pleased to inform you that your manuscript has been deemed suitable for publication in PLOS ONE. Congratulations! Your manuscript is now with our production department. 

Kind regards, 

on behalf of

Professor Anwar Khitab 

Academic Editor

PLOS ONE